# *OsbHLHq11*, the Basic Helix-Loop-Helix Transcription Factor, Involved in Regulation of Chlorophyll Content in Rice

**DOI:** 10.3390/biology11071000

**Published:** 2022-07-01

**Authors:** Yoon-Hee Jang, Jae-Ryoung Park, Eun-Gyeong Kim, Kyung-Min Kim

**Affiliations:** 1Department of Applied Biosciences, Graduate School, Kyungpook National University, Daegu 41566, Korea; uniunnie@naver.com (Y.-H.J.); dkqkxk632@naver.com (E.-G.K.); 2Crop Breeding Division, Rural Development Administration, National Institute of Crop Science, Wanju 55365, Korea; icd0192@korea.kr

**Keywords:** CIELAB, photosynthesis, QTL, SPAD, yield

## Abstract

**Simple Summary:**

R-ice is one of the world’s most important staples; a growing population and declining rates of growth in rice yields will present significant challenges ahead. After the heading stage, the photosynthetic ability of the flag leaf has a great effect on the yield of rice, and this ability can be evaluated by leaf color, chlorophyll content, quantum yield, etc. Our purpose was to screen candidate genes that affect photosynthetic efficiency through QTL mapping analysis and predict their function through protein interaction and homology sequence analysis. The results suggest that *OsbHLHq11* may be involved in chlorophyll accumulation and enhancing photosynthetic efficiency, which may lead to high yields.

**Abstract:**

Photosynthesis is an important factor in determining the yield of rice. In particular, the size and efficiency of the photosynthetic system after the heading has a great impact on the yield. Research related to high-efficiency photosynthesis is essential to meet the growing demands of crops for the growing population. Chlorophyll is a key molecule in photosynthesis, a pigment that acts as an antenna to absorb light energy. Improvement of chlorophyll content characteristics has been emphasized in rice breeding for several decades. It is expected that an increase in chlorophyll content may increase photosynthetic efficiency, and understanding the genetic basis involved is important. In this study, we measured leaf color (CIELAB), chlorophyll content (SPAD), and chlorophyll fluorescence, and quantitative trait loci (QTL) mapping was performed using 120 Cheongcheong/Nagdong double haploid (CNDH) line after the heading date. A major QTL related to chlorophyll content was detected in the RM26981-RM287 region of chromosome 11. *OsbHLHq11* was finally selected through screening of genes related to chlorophyll content in the RM26981-RM287 region. The relative expression level of the gene of *OsbHLHq11* was highly expressed in cultivars with low chlorophyll content, and is expected to have a similar function to BHLH62 of the *Gramineae* genus. *OsbHLHq11* is expected to increase photosynthetic efficiency by being involved in the chlorophyll content, and is expected to be utilized as a new genetic resource for breeding high-yield rice.

## 1. Introduction

One of the most widely grown crops, *Oryza sativa L.*, is the staple food of about 3.5 billion people worldwide [1]. The world population is expected to increase from 6.13 billion in 2001 to 8.27 billion in 2030 [2]. This will require yields of key crops to increase by more than 70% over the next 30 years to sustain human requirements [3]. However, since 2000, the global rice growth rate has decreased, which is less than the world rice consumption [4]. Increases in rice production driven by the Green Revolution have supported global demand for many years, but the increase in rice yields driven by genetic advances in breeding has now dropped to less than 1% [5]. Efforts are needed to increase both genetic gain and the efficiency of agricultural practices to meet this burgeoning demand [6]. Yield growth must be achieved without increasing the use of arable land or adversely affecting the environment [7]. In particular, high-input agricultural systems incur significant economic and environmental costs [8], so it is essential to develop new varieties of major crops to increase yield while protecting the environment [9].

Photosynthesis is emerging as a key pathway to increase the potential genetic yield of our major crops [10]. Photosynthesis is the most important energy source for plant growth, and crop yield is determined by the aggregate rate of photosynthesis over the growing season [11]. In fact, more than 90% of crop biomass is derived from photosynthesis, so many crop scientists believe that enhancing photosynthesis at the single leaf level will increase yields [12]. Whether an increase in leaf photosynthesis increases yield is debatable, but recent studies have shown that the growth rate around the harvest stage of rice is critically related to the final yield of rice [13]. From previous studies, it can be predicted that improvements in flag leaf photosynthesis will help increase the potential of future crop yields [14]. While the improvement of the canopy structure according to the length, width, and erection of leaves played an important role in traditional crop breeding, the improvement of leaf photosynthetic properties was slight [15]. Recently, more attention has been paid to genetic studies of rice photosynthesis and rice breeding for the physiological properties related to high-efficiency photosynthesis, and novel biotechnological approaches to promote photosynthesis in rice plants are expected to increase crop yields by up to 27% [16,17].

Chlorophyll content is one of the most important physiological properties of plants because it is closely related to the photosynthetic and yield potential of leaves [18]. Chlorophyll is called the central pigment of photosynthesis because it can accommodate light absorbed by other pigments through photosynthesis, and is an important part of the plant’s photosynthetic apparatus for harvesting light and producing biochemical energy for use within the Calvin ycle [19,20]. Chlorophyll is a complex organic molecule containing a central magnesium ion and a pyrrole derivative that forms a complex porphyrin system, essential for the thylakoid reaction of photosynthesis [21]. Therefore, the chlorophyll content of the leaf is closely related to the photosynthetic function of the plant and represents the physiological state of the plant [22]. Chlorophyll affects the photosynthetic capacity of plants such as light blocking, penetration, and conversion, which is also linked to crop productivity [23]. In practice, the loss of photosynthesis is mainly due to the loss of functional units of photosynthesis or a decrease in the total number of chloroplasts [24]. Improvement of chlorophyll content characteristics has been emphasized in rice breeding for decades, and new high-yielding rice cultivars have been reported to have higher chlorophyll content and leaf photosynthetic rates than previously released low yield cultivars [25].

Recent advances in molecular genetic mapping and marker generation for rice have made possible the genetic mapping of quantitative trait loci (QTL) to photosynthetic-associated traits [26]. Through QTL analysis, it is possible to comprehensively analyze the genetic relationship between morphological traits and agronomic traits. QTLs can also be used to determine the most probable candidates among the various genes that control traits [27]. Previous studies have shown that genes controlling chlorophyll content constitute a multi-allelic system representing QTLs [28]. However, none of these QTLs have been accurately mapped to Mendelian factors or characterized as contributing to leaf photosynthesis [29]. Therefore, in this study, the chlorophyll content, leaf color, and quantum yield of photosystem II (F_v_/F_m_) related to photosynthetic efficiency were analyzed using the leaves after the outgrowth, and photosynthetic efficiency and related genes were identified through QTL mapping. Thereafter, the gene expected to be most related to photosynthetic efficiency among the candidate genes was selected through relative gene expression. In addition, by analyzing the protein homology of the selected genes, we propose a gene related to the photosynthetic efficiency of rice, and the results are expected to be effectively used in the development of rice cultivars.

## 2. Materials and Methods

### 2.1. Plant Materials and Field Design

In this study, 120 Cheongcheong/Nagdong doubled haploid (CNDH) line developed by another culture of F_1_ derived from a cross between Cheongcheong (*Oryza sative* spp. *indica* cv. Cheongcheong) and Nagdong (*Oryza sative* spp. *japonica* cv. Nagdong) was used as a plant material for QTL mapping. The CNDH line has been developed in the field for more than 10 years, and the agricultural traits are perfectly fixed. The CNDH group was obtained from Prof. Kim Kyung-min of the Plant Molecular Breeding Laboratory at Kyungpook National University, and was developed in the field of Kyungpook National University (36°6′41.54″ N, 128°38′26.17″ E) in 2021. Seeds were sterilized using 500 μL of spotack (Hankook Samgong, Seoul, Korea) per 1 L of water, and sown in plastic trays after soaked in dark conditions at 25 °C for 4 days. The sown seeds were cultivated in a greenhouse, and 30 days after sowing, the seedlings were transplanted to the field of Kyungpook National University in Gunwi, and the planting interval was 30 × 15 cm. Fertilizers were applied to the fields at the rate of N-P_2_O_5_-K_2_O = 9-4.5-5.7 kg/10a according to the Rural Development Administration’s standard application rate. After that, leaf color, chlorophyll content, and F_v_/F_m_ were measured at the 0, 15, 30, 45 days after heading date.

### 2.2. Measurement of Leaf Color, Chlorophyll Content, and F_v_/F_m_

In order to evaluate the photosynthetic efficiency after heading of the CNDH 120 line, CIELAB L*, a*, b* values, chlorophyll content, and F_v_/F_m_ were measured. The measurement site was based on the middle part of the branch leaf. The leaf color was measured using a color meter (TES-135A, TES Co., Ltd., Taipei, Taiwan) to measure the CIELAB value, and white calibration was performed before measurement for accurate measurement. After setting the color meter to CIELAB (set to L*a*b*, ΔE*), the CIELAB value was measured by vertically adhering to the flag leaf. The CIELAB value was defined by the International Commission on Illumination (CIE) and is used to measure chromaticity diagrams at the same level as human visual perception. L* value indicates lightness from black (0) to white (100), a* value indicates green (−) to red (+), and b* value indicates blue (−) to yellow (+) value. RGB is converted to XYZ color space to obtain L*a*b* values [30]. Hue angle represents the angle on the 360° color wheel, and 0, 90, 180, and 270° represent the shades of red, yellow, green, and blue, respectively. Chroma (C*ab) represents the hypotenuse of a right triangle created by connecting the points (0,0), (a*, b*), and (a*, 0), and the values of hue angle and C*ab are calculated by the same formula. Hue angle (°) = arctan b*/a*, C*ab = (a*^2^ + b*^2^)*1/2 [31]. Chlorophyll content was measured using a portable chlorophyll meter (SPAD-502, Minolta Camera Co., Ltd., Osaka, Japan), and the absorbance of leaves was measured in two wavelength ranges (wavelength sensitive to chlorophyll; 650 mm, wavelength insensitive to chlorophyll; 940 mm). The relative level of chlorophyll present was measured and expressed as the SPAD value. F_v_/F_m_ was measured using a chlorophyll fluorometer (FluorPen FP 100, Photon System Instruments, Czech Republic). Before measurement, the middle part of the leaf was dark-treated using a leaf-clip for 30 min, then the fluorometer was attached to the leaf-clip vertically and the F_v_/F_m_ value was measured after removing the dark-treated plate of leaf-clip. The derived parameters were averaged using FluorPen 1.0 software (Photon System Instruments, Czech Republic) based on the formula reported by Strasser et al. [32]. All traits were measured five times for each CNDH line.

### 2.3. Response Surface Methodology (RSM) Analysis

To investigate the effect of CIELAB value on chlorophyll content, RSM analysis was performed focusing on CIEa* and CIEb* values. Minitab 17 (Minitab Inc., State College, PA, USA) was used for RSM analysis.

### 2.4. Construction of Genetic Map and QTL Analysis

For QTL mapping, Windows QTL Cartographer 2.5 program was used, and a genetic map with an average interval of 10.6 cMbetween markers, created using Mapmaker version 3.0 using 222 SSR markers, was used in this study. The program requires factors such as the genetic distance between each marker, the name of the marker, the number of chromosomes, genotyping data, and the target trait value. Composite interval mapping (CIM) method was used, and LOD scare was set to 2.5 or higher and QTL analysis was performed.

### 2.5. Gene Information Analysis

Genes related to chlorophyll content were searched for in the detected QTL region with an LOD score of 2.5 or higher. Rapdb (https://rapdb.dna.afrc.go.jp/ accessed on 1 November 2021) was used to confirm the marker position on the chromosome, and the search for open reading frames (ORF) corresponding to the marker region was conducted by RiceXpro (https://ricexpro.dna.afrc.go.jp/ accessed on 15 June 2022, verson 3.0). Chlorophyll-related genes were selected by analyzing the functions of all discovered ORFs, the genes were identified by Simple Modular Architecture Research Tool (SMART, http://smart.embl-heidelberg.de/ accessed on 10 February 2022), and ExPASy (https://www.expasy.org accessed on 21 February 2022) was used to identify protein functions and interactions. In addition, homology multiple sequences were analyzed using NCBI (http://www.ncbi.nim.nih.gov accessed on 6 April 2022) and Molecular Evolutionary Genetics Analysis software (MEGA X, PA, USA). The phylogenetic tree was created by applying the neighbor-joining method and the bootstrap method (*n* = 1000).

### 2.6. RNA Extraction

Total RNA was extracted from the leaves of the Cheongcheong, Nagdong, and CNDH 120 line at 0 DAH using the RNeasy plant mini kit (Qiagen, Cat. No. 74903, Hilden, Germany). All instruments used for RNA extraction were sterilized, and samples for RNA extraction were maintained at 4 °C during the extraction process. Then, 30 mg of rice leaves stored at −80 °C were frozen using liquid nitrogen and ground until fine powder was obtained using a mortar. After the ground sample was placed in a 2 mL e-tube, 450 µL of RLT buffer containing β-mercaptoethanol was added and vortexed vigorously. After transferring the lysate to the QIA shredder spin column located in a 2 mL collection tube, the vortexed solution was centrifuged at 13,000 rpm for 2 min. The supernatant from the flow-through was transferred to a 1.5 mL e-tube and mixed by pipetting immediately after adding 500 µL of ethanol. Then, 650 µL of sample was transferred to a RNeasy Mini spin column placed in a 2 mL collection tube and centrifuged at 13,000 rpm for 15 s. The flow-through was discarded, 700 µL of RW1 buffer was added, and the sample was centrifuged at 13,000 rpm for 15 s. Then, after 500 µL of RPE buffer was added, the sample was centrifuged at 13,000 rpm for 15 s. After 500 µL of RPE buffer was added once more, the sample was centrifuged at 13,000 rpm for 2 min. In addition, centrifugation was further performed at 13,000 rpm for 1 min to completely dry the column membrane. The RNeasy spin column was placed in a 1.5 mL collection tube, 50 µL of RNase-free water was added in the middle of the spin column membrane, and centrifugation was performed at 13,000 rpm for 1 min. The extracted RNA was quantified using an ultramicro spectrophotometer (Nanodrop, ND-2000, Santa Clara, CA, USA), and 1 µg of the extracted total RNA was used as a template for cDNA synthesis.

### 2.7. Analysis of Relative Expression Level

cDNA used as a quantitative real-time PCR template for analysis of relative expression level was synthesized using UltraScript 2.0 cDNA Synthesis Kit (PCR Biosystems Wayne, PA, USA). For cDNA synthesis, RNA 1 µg, 5× cDNA Synthesis Mix 4.0 µL, and UltraScript 2.0 for cDNA Synthesis (with RNase inhibitor) 1.0 µL were added, and the final volume was adjusted to 20 µL using RNase-free water. Then, that solution was incubated at 50 °C for 30 min. The reaction solution used for quantitative real-time PCR consisted of 1 μL of cDNA, 10 μL of 2× Real-Time PCR Master Mix, 1 μL of forward primer (20 pmol/μL), 1 μL of reverse primer (20 pmol/μL), and a final volume of 20 μL was adjusted using ddH_2_O. Quantitative real-time PCR was performed using Eco Real-Time PCR System. PCR conditions were performed at 95 °C for 15 min (Holding stage), at 95 °C for 20 s, at 60 °C for 40 s (cycling stage, repeated 40 cycles), at 95 °C for 15 s, 60 °C for in 1 min, and at 95 °C for 15 s (melt curve). The comparative 2^−ΔΔCt^ method, which is calculated based on the difference in Ct values, was used to calculate the relative expression level [33], and *OsActin*, a housekeeping gene, was used as a control. Each reaction was performed three times to calculate the mean and standard deviation.

### 2.8. Statistical Analysis

All experiments were repeated three times for each sample, and statistical analysis to calculate the mean and standard deviation was performed using the SPSS program (IMMSPSS Statistics, version 22, IBMSPSS Statistics, version 22, Redmond, WC, USA).

## 3. Results

### 3.1. Measurement of Leaf Color, Chlorophyll Content, and F_v_/F_m_ after Heading

Leaf color, chlorophyll content, and F_v_/F_m_ were measured after 0, 15, 30, and 45 days after the heading date (DAH) of CNDH 120 line. During all measurement periods, parents Cheongcheong and Nagdong had higher CIEL* values than CNDH 120 line, and both Cheongcheong, Nagdong, and CNDH 120 line increased CIEL* values over time after the heading date. CIEa* had a negative value for all of Cheongcheong, Nagdong, and CNDH 120 line, and a negative value means that it is closer to green than red. In all of the Cheongcheong, Nagdong, and CNDH 120 line, the CIEa* values gradually increased after the heading date and sharply increased on the 45 DAH. The averages of Cheongcheong, Nagdong, and CNDH 120 line were similar until 30 DAH, but on the 45 DAH, Cheongcheong and Nagdong showed a higher CIEa* value than that of the CNDH 120 line. CIEb* also showed a gradually increasing value after the heading date, and showed a large value in the order of Nagdong, CNDH 120 line, and Cheongcheong until 30 DAH. Hue angle and C*ab were calculated using the coordinates of CIEa* and CIEb*. Hue angle showed similar values for Cheongcheong, Nagdong, and CNDH 120 line. At 15 DAH, Cheongcheong had a smaller C*ab value than that of Nagdong and CNDH 120 line, and in 45 DAH, the C*ab value of CNDH 120 line was smaller than that of Cheongcheong and Nagdong. In Cheongcheong, Nagdong, and CNDH 120 line, the chlorophyll content tends to decrease after the heading date, and it can be seen that the chlorophyll content decreases rapidly in 45 DAH. The F_v_/F_m_ also tended to decrease after the heading date in Cheongcheong, Nagdong, and CNDH 120 line, and in 45 DAH, Nagdong showed a higher F_v_/F_m_ value than Cheongcheong and CNDH 120 line (Table 1).

The frequency distribution table of CIELAB values, chlorophyll content, and F_v_/F_m_ of Cheongcheong, Nagdong, and CNDH 120 line showed continuous variation close to normal distribution at 0, 15, 30, and 45 DAH. This means that CIELAB value, chlorophyll content, and F_v_/F_m_ are quantitative traits were confirmed to be suitable for QTL analysis (Figure 1). As a result of analyzing the correlation between the chlorophyll content and related traits, the chlorophyll content showed a positive correlation with the yield (0 DAH *r^2^* = 0.246, 15 DAH *r^2^* = 0.303, 30 DAH *r^2^* = 0.273, 45 DAH *r^2^* = 0.032), hue angle (0 DAH *r^2^* = 0.486, 15 DAH *r^2^* = 0.496, 30 DAH *r^2^* = 0.573, 45 DAH *r^2^* = 0.644), F_v_/F_m_ (0 DAH *r^2^* = –0.016, 15 DAH *r^2^* = 0.017, 30 DAH *r^2^* = 0.479, 45 DAH *r^2^* = 0.474) and a negative correlation with CIEa* (0 DAH *r^2^* = 0.041, 15 DAH *r^2^* = –0.243, 30 DAH *r^2^* = –0.379, 45 DAH *r^2^* = –0.674), CIEb* (0 DAH *r^2^* = –0.596, 15 DAH *r^2^* = –0.270, 30 DAH *r^2^* = –0.409, 45 DAH *r^2^* = –0.668), C*ab (0 DAH *r^2^* = –0.499, 15 DAH *r^2^* = –0.078, 30 DAH *r^2^* = –0.211, 45 DAH *r^2^* = –0.481), (Table 2).

Response surface methodology (RSM) analysis was performed for CIEa*, CEIb*, and chlorophyll content to find out how the CIELAB values, CIEa* and CEIb*, affect the chlorophyll content. Chlorophyll content was highest when CIEa* value was –35 and CIEb* value was 20 at 0 DAH, CIEa* value was –37 and CIEb* value was 20 at 15 DAH, CIEa* value was –28 and CIEb* value was 25 at 30 DAH, and CIEa* value was –28 and CIEb* value was 31 at 45 DAH. Overall, the smaller the CIEa* and CIEb* values, the higher the chlorophyll content tended to be (Figure 2).

### 3.2. Analysis of QTLs Associated with Leaf Color, Chlorophyll Content, and F_v_/F_m_

A genetic map of the CNDH line for QTL analysis was generated using the 778 SSR marker. In total, 423 SSR markers showed polymorphism in the parent Cheongcheong and Nagdong, and among them, 143 SSR markers showed co-dominance in both Cheongcheong and Nagdong and were selected to create a genetic map of the CNDH line. The average distance between the markers used to generate the genetic map was 10.6 cM and 19–50 markers were used per chromosome. To analyze the QTL related to the chlorophyll content, Windows QTL Cartographer 2.5 was used, and the Composite Interval Mapping (CIM) method was used. Logarithm of the odds (LOD) value was set to 2.5, and QTLs were named according to the method suggested by McCouch et al. [34].

As a result of QTL mapping for chlorophyll content and related traits, a total of 25 QTLs were found in 8 chromosomes. qHa2, qHa6, qCab8, qCc7, and qCc11 were detected at 0 DAH; qCb3, qCb7, qCb1, qHa3, qHa7, qCab6, qCc1, qCc6, qCc11-1, and qCc11-2 were detected at 15 DAH; qCa6, qCa8, qHa12, and qCab1 were detected at 30 DAH; and qCa3, qCa3-1, qCa12, qCc11-3, and qPqy3 were detected at 45 DAH (Figure 3). qCa is a QTL associated with CIEa*. qCa6 and qCa8 were detected at 30 DAH, and qCa3, qCa3-1, and qCa12 were detected at 45 DAH. qCa6 was detected in the RM20158-RM20017 region of chromosome 6, and the LOD score was 3.45. The proportion of evaluated phenotype variation (*R*^2^) contributing to this QTL was 30%, derived from the Cheongcheong allele. qCa8 was detected in the RM1148-RM22197 region of chromosome 8, and the LOD score was 4.41. *R*^2^ was 26%, derived from the Cheongcheong allele. qCa3 was detected in the RM2334-RM3525 region of chromosome 3, and the LOD score was 2.63. *R*^2^ was 32%, derived from the Nagdong allele. qCa3-1 was derived from the RM15927-RM16146 region of chromosome 3, and LOD score was 3.82. *R*^2^ was 26%, derived from the Cheongcheong allele. qCa12 was derived from the RM1246-RM1261 region of chromosome 12, and LOD score was 3.51. *R*^2^ was 27%, derived from the Cheongcheong allele. qCb is a QTL related to CIEb*, and qCb1, qCb3 and qCb7 were detected at 15 DAH. qCb1 was detected in the RM11694-RM3530 region of chromosome 1, and the LOD value was 2.92. *R*^2^ was 20%, derived from the Cheongcheong allele. qCb3 was found in the RM3523-RM1221 region of chromosome 3, and the LOD value was 2.50. *R*^2^ was 20% derived from the Cheongcheong allele. qCb7 was detected in the RM20924-RM20967 region of chromosome 7, and the LOD value was 3.70. *R*^2^ was 27%, derived from the Cheongcheong allele. qHa is a QTL related to hue angle, and qHa2 and qH6 were detected at 0 DAH, qHa3 and qHa7 were detected at 15 DAH, and qHa12 was detected at 30 DAH. qHa2 was detected in the RM12532-RM12662 region of chromosome 2, and the LOD value was 3.63. *R*^2^ was 26%, derived from the Cheongcheong allele. qHa6 was detected in the RM217-RM588 region of chromosome 6, and the LOD value was 3.71. *R*^2^ was 29% derived from the Nagdong allele. qHa3 was detected in the RM2334-RM1221 region of chromosome 3, and the LOD value was 4.38. *R*^2^ was 26% derived from the Cheongcheong allele. qHa7 was detected in the RM20924-RM20967 region of chromosome 7, and the LOD value was 2.75. *R*^2^ was 23% derived from Nagdong allele. qHa12 was detected in the RM1246-RM1261 region of chromosome 12, and the LOD value was 2.80. *R*^2^ was 24%, derived from the Nagdong allele. qCab is a QTL related to C*ab, with qCab8 at 0 DAH, qCab15 at 15 DAH, and qCab1 at 30 DAH. qCab8 was detected in the RM23191-RM44 region of chromosome 8, and the LOD value was 3.19. *R*^2^ was 24%, derived from the Nagdong allele. qCab6 was detected in the RM20017-RM217 region of chromosome 6, and the LOD value was 3.10. *R*^2^ was 28% derived from the Nagdong allele. qCab1 was detected in the RM11694-RM1297 region of chromosome 1, and the LOD value was 3.10. *R*^2^ was 21%, derived from the Cheongcheong allele. qCc is a QTL related to chlorophyll content. qCc7 and qCc11 were detected at 0 DAH; qCc1, qCc6, qCc11-1, qCc11-2, and qCc12 were detected at 15 DAH; and qCc11-3 was detected at 45 DAH. qCc7 was detected in the RM21105-RM21582 region of chromosome 7, and the LOD value was 3.84. *R*^2^ was 36%, derived Nagdong allele. qCc11 was detected in the RM26981-RM287 region of chromosome 11, and the LOD value was 2.95. The *R*^2^ value was 31%, derived from Cheongcheong allele. qCc1 was detected in the RM11605-RM3530 region of chromosome 1, and the LOD value was 2.50. *R*^2^ was 36%, derived from the Cheongcheong allele. qCc6 was detected in the RM20196-RM20092 region of chromosome 6, and the LOD value was 2.78. *R*^2^ was 32% derived from Nagdong allele. qCc11-1 was detected in the RM26981-RM287 region of chromosome 11, and the LOD value was 4.78. *R*^2^ was 33%, derived from Cheongcheong allele. qCc11-2 was detected in the RM287-RM27161 region of chromosome 11, and the LOD value was 2.65. *R*^2^ was 33%, derived from Cheongcheong allele. qCc12 was detected in the RM287-RM27161 region of chromosome 12, and the LOD value was 2.65. *R*^2^ was 33%, derived from the Nagdong allele. qCc11-3 was detected in the RM167-RM3428 region of chromosome 11, and the LOD value was 3.12. *R*^2^ was 26%, derived from the Nagdong allele. qPqy is a QTL related to photosynthetic efficiency, and qPqy3 was detected. qPqy3 was detected in the RM15927-RM15904 region of chromosome 3, and the LOD value was 3.45. *R*^2^ was 25%, derived from Nagdong allele (Table 3).

### 3.3. Search for Chlorophyll-Related Gene Based on QTL Mapping

As a result of QTL mapping for chlorophyll content and related traits at 0, 15, 30, and 45 days after heading date, QTLs with overlapping regions were detected in RM11694-RM1297 on chromosome 1, RM2334-RM15904 on chromosome 3, RM20924-RM2182 on chromosome 7, and RM26981-RM287 on chromosome 11. The RM26981-RM287 region of chromosome 11 was detected with the highest LOD value of 4.78 among the searched QTLs related to chlorophyll content. Therefore, in this study, candidate genes related to chlorophyll content were searched by focusing on the RM26981-RM287 region. The marker interval of RM26981-RM287 on chromosome 11 was 19.9 cM and this SSR marker was analyzed by NCBI to screen for open reading frames (ORFs) related to chlorophyll content and classified according to molecular function (Appendix A). As a result, 26 OFRs related to chlorophyll content were detected in the RM26981-RM287 region, and each ORF has various functions such as transporter, transcriptional factor, signaling, RNA binding, protein binding, nucleotide binding, kinase, DNA binding, and catalytic activity (Figure 4). The candidate genes related to transporter function were twin-arginine translocation protein, prenyltransferase domain containing protein, PDR11 ABC transporter, and ABCF-type protein. The candidate genes related to signaling were phytosulfokine family protein and PAP fibrillin family protein. The candidate gene related to RNA binding function was CRS1/YhbY domain containing protein, 50S ribosomal protein L4. The candidate genes related to protein binding function include Thioredoxin domain 2 containing protein. The candidate genes related to nucleotide binding function include root hair defective 3 protein and 14-3-3 protein. The candidate genes related to DNA binding function include BSD domain containing protein, HMG-I and HMG-Y domain containing protein, Basic helix-loop-helix dimerisation region bHLH domain containing protein. The candidate genes related to catalytic activity function were glutamyl-tRNA amidotransferase, terpene synthase family protein, and short chain alcohol dehydrogenase.

### 3.4. Relative Expression Levels of Chlorophyll Content Related Genes

In the RM26981-RM287 region of chromosome 11, 26 chlorophyll content-related genes were screened. In order to compare the expression level of candidate genes according to the chlorophyll content, CNDH11 and CNDH115, which are lines with high chlorophyll content, and CNDH50-2 and CNDH52-2, with low chlorophyll content, were used (Figure 5). The flag leaf of 0 DAH, when the average of chlorophyll content of CNDH 120 line is highest, was used for comparison of relative expression level. Three of the 26 candidate genes showed differences between CNDH11 and CNDH115 with high chlorophyll content and CNDH50-2 and CNDH52-2 with low chlorophyll content. In *LOC_Os11g34210*, *LOC_Os11g37130* and *LOC_Os11g38870*, the relative expression levels of CNDH11 and CNDH115 were lower than those of CNDH50-2 and CNDH52-2, and this difference was significant at the 1% level. For *LOC_Os11g34210* and *LOC_Os11g38870*, the relative expression levels of CNDH11 and CNDH115, a CNDH line with a high chlorophyll content, were not different, and the relative expression levels of CNDH50-2 and CNDH52-2, a CNDH line with a low chlorophyll content, were also not different. However, since the relative expression levels of CNDH11 and CNDH115 were different for *LOC_Os11g37130*, it is difficult to say that the relative expression level of *LOC_Os11g37130* differs depending on the chlorophyll content (Figure 6). Therefore, *LOC_Os11g34210* and *LOC_Os11g38870* were selected as candidate genes related to chlorophyll content. Among them, *LOC_Os11g38870* (*OsbHLHq11*) with a CDS size of 837 bp and a basic helix-loop-helix (bHLH) domain involved in chlorophyll biosynthesis and phytochrome signal transduction was finally selected.

### 3.5. Analysis of Homology Sequence and Protein Interaction

As a result of BLAST analysis in NCBI, it was found that OsbHLHq11 has a sequence very similar to that of BHLH062 protein. Using the sequence of OsbHLHq11, the genetic similarity of BHLH062 of *Oryza sativa* L., *Oryza brachyantha*, *Lolium rigidum*, *Brachypodium distachyon*, *Triticum aestivum*, *Hordeum vulgare*, and *Setaria italica* belonging to the *Gramineae* family was analyzed. OsbHLHq11 showed the most similar genetic similarity (identity 87%, similarity 93%) to the BHLH062-like isoform ×1 protein of *Oryza brachyantha*, and belonged to the same group in the phylogenetic tree. BHLH062-like isoform X2 protein from *Oryza sativa* L. (identity 99%, similarity 92%), BHLH062-like protein from *Lolium rigidum* (identity 70%, similarity 92%), BHLH062-like protein from *Triticum aestivum* (identity 66%, similarity 92%), BHLH062 protein from *Brachypodium distachyon* (identity 72%, similarity 90%), and BHLH062 protein from *Setaria italica* (identity 70%, similarity 89%) are in the order of the highest genetic similarity. The genetic similarity of BHLH062-like isoform ×1 protein from *Hordeum vulgare* showed the greatest difference (identity 68%, similarity 89%) with OsbHLHq11 (Figure 7B,D). When protein interactions were analyzed using the domain of OsbHLHq11, it was found to interact with 10 different proteins, NAS1, IRO2, OsHRZ1, OsJ_12589, OsJ_32857, OsJ_17424, OS07T0659900-01, OS05T0551000-01, OS04T0444800-02, and OS04T0381700-00 (Figure 7C).

## 4. Discussion

Rice is a staple food for nearly half of the world’s population, and increasing rice production to meet growing demand from an ever-growing population faces many challenges [35]. There is an urgent need to develop efficient cultivars that require less water, labor, nitrogen, and pesticides, and improving photosynthetic performance and efficiency may be the most promising approach to solving these problems [8,36]. A number of studies have shown that increased photosynthetic capacity actually increases yield under field conditions for rice [11]. Cao et al. evaluated the photosynthetic efficiency of rice germplasm and found that the difference in photosynthetic rate between rice cultivars was significant. They argued that high-yielding cultivars could be developed by selecting cultivars with a high photosynthetic efficiency [37]. Additionally, Huang et al. suggested that higher RUE and grain yield could be achieved by improving leaf photosynthetic properties, including chlorophyll a content, F_v_/F_m_, and Rubisco content of hybrid rice [38]. Chlorophyll concentration is routinely used as an index to estimate photosynthetic efficiency [39]. Chlorophyll a (Chl a) and chlorophyll b (Chl b) are essential for the primary reactions of photosynthesis [40]. Chl a is essential for photochemistry and chl b is required to stabilize the main light-harvesting chlorophyll binding protein [41]. On the other hand, chlorophyll and its derivatives, as strong photosensitizers, cause reactive oxygen species when excessively present. To maintain healthy growth, the metabolism of chlorophyll is highly regulated during plant development [42]. Methods for measuring chlorophyll content include a method based on absorption of light by a water-soluble acetone extract of chlorophyll, and a method based on the absorbance or reflectance of light of a specific wavelength by an intact leaf using a portable chlorophyll meter [43]. However, these destructive methods based on laboratory procedures such as acetone-ethanol extraction, spectrophotometry, and high-performance liquid chromatography are time-consuming, expensive, and not suitable for high-throughput analysis [44]. Chlorophyll mainly determines the color of plant leaves, which indicates the nutritional and health status of plants, and it has been demonstrated that plant nutrient levels, water availability, plant diseases, and aging have a significant effect on plant leaf color [45]. Chlorophyll absorbs 70–90% of light between 430–450 nm (blue) and 640–660 nm (red) wavelengths, while lower absorption occurs at 550 nm (green) wavelengths [46]. As a result, the reflectance and transmittance in the visible light region are small, but reach the maximum in green, which represents the green color of leaves. Under field conditions, the color of plant leaves can be qualitatively evaluated and differentiated through color charts such as Munsell Color Chart for Plant Tissues, Globe Plant Color Chart, and Leaf Color Chart [47]. This is a simple and inexpensive technique to obtain the color of plant leaves, and it depends on the human eye and perception of sunlight conditions [48]. However, the human eye can visually misread the color of the LCC and cause frequent fading of the color chart, which can lead to improper application [49]. We used CIELAB histograms to effectively extract the color characteristics of each area in situ. This can be used to distinguish the color level of rice leaves and to evaluate the chlorophyll content. Chlorophyll fluorescence is one of the most widely used techniques in plant physiology because users can easily obtain detailed information about the status of photosystem II (PSII) at a relatively low cost [50]. Various computational parameters such as F_v_/F_m_ have been intensively adopted to diagnose the physiological state of plants [51]. Chlorophyll fluorescence analysis has been shown to be a non-invasive, robust, and reliable method for assessing changes in the phenomenological and biophysical expression of photosystem II in different species for different environmental conditions, such as light intensity [52]. Chlorophyll fluorescence measurements are based on capturing and measuring the light emitted by chlorophyll again while the chlorophyll molecules return from an excited state to a non-excited state [3]. When there are more chlorophyll molecules that cannot transmit energy to the reaction center, the value of F_o_ increases. In stressed plants, it usually also shows a decrease in F_m_. These two values are large when the chlorophyll content is high. The value obtained by subtracting F_o_ from F_m_ is called maximum variable fluorescence (F_v_), and the value obtained by dividing F_v_ by F_m_ means the maximum quantum yield of PSII for the photochemical reaction [53]. In other words, this value indicates the potential for photosynthesis of plant leaves [54]. So, the efficiency of the photochemical reaction can be estimated by measuring the quantum yield of PSII. Therefore, to find genes related to photosynthetic efficiency in the CNDH line, we measured the chlorophyll content, the color of rice leaves that change due to the chlorophyll content, and quantum yield of photosystem II, which represents the efficiency of use of light energy captured by the chlorophyll molecule in the CNDH line, and the correlation between them was analyzed. Flag leaf was used for the measurement of various traits. The photosynthetic capacity of the flag leaf plays an important role in determining crop yield [55]. During the grain filling period, photosynthesis of rice plants contributes 60–100% of the final grain carbon content [56]. To achieve a potential yield, the metabolic activity within the grain must match the maximum activity of the source leaf, and, in fact, high-yielding cultivars have leaves capable of maintaining photosynthetic activity throughout the grain filling period [57]. Therefore, it is very important to genetically analyze the morphology and physiological characteristics of functional leaves, especially flag leaf, in rice improvement [58].

The CIELAB values after heading, the hue angle, and C*ab derived from the CIEa* and CIEb* values showed a normal distribution, and the chlorophyll content and F_v_/F_m_ values also showed a normal distribution. This indicates that the phenotypic distribution of the measured traits can be defined as a continuous trait, the variation of these traits being controlled by the segregation of a number of loci, often referred to as QTL [59,60]. Chlorophyll content is highly correlated with CIELAB values, which are significant at the 1% level. Chlorophyll content shows a negative correlation with CIEa* and CIEb* values, that is, when CIEa* values are closer to green (−) than red (+) and CIEb* values are closer to blue (−) rather than yellow (+), the chlorophyll content is measured to be higher. The chlorophyll content has no correlation with the F_v_/F_m_ value at 0 DAH and 15 DAH, but shows a high correlation at 30 DAH and 45 DAH. It seems that at the beginning of heading, the F_v_/F_m_ values are similar because the plants are healthy, but the amount of chlorophyll fluorescence emission varies depending on the CNDH line due to the stress that the plants receive as the plants age after heading [61].

Response surface methodology (RSM) analysis shows how CIELAB color values CIEa* and CIEb* affect chlorophyll content [62]. RSM is an empirical statistical method suitable for multi-factor experimental design because it considers the interaction effect between process variables [63]. Since this is usually applied to evaluate model-generated multiple regression equations, the optimal value of each independent factor can be determined to maximize the response variable. RSM has been widely used in biotechnology processes and has succeeded in optimizing culture conditions for enzyme immobilization, protein extraction, fermentation, and biofuel production [64]. Just as the CIEa* and CIEb* values had a negative correlation with the chlorophyll content, the RSM analysis also shows that the lower the CIEa* and CIEb* values after heading, the higher the chlorophyll content.

As a result of QTL mapping of chlorophyll content, leaf color, and quantum yield of PSII at 0, 15, 30, and 45 days after heading date using the CNDH line, the QTLs related to the CIELAB value is chromosome 1, 2, 3, 6, 7, 8, and 12; QTLs related to chlorophyll content were detected in chromosomes 1, 6, 7, 11, 12; and QTLs related to quantum yield of PSII were detected in chromosome 3. In particular, three QTLs related to chlorophyll content were detected on chromosome 11, and QTLs related to chlorophyll content on 0 DAH and 15 DAH were detected in the RM26981-RM287 region. With the rapid development of molecular markers and the application of high-resolution connectivity maps in recent years, some QTLs related to chlorophyll content have been identified. QTLs were detected for SPAD values on chromosome 4 using rice population derived from a cross between the Sasanishiki, Japonica cultivars, and Habataki, Indica cultivars with high yield [29]. In Indica/Japonica hybrid rice, QTLs for leaf chlorophyll content were detected in the short arm of chromosome 6 and long arm of chromosome 9 after 25 days of flowering [65]. A double haploid (DH) population was developed from an Indica/Japonica (Zhenshan 97/Wuyujing 2) cross and a DH line was backcrossed to detect a total of 60 QTLs related to chlorophyll content in the two backcross populations. This QTL is distributed in 10 rice chromosomes except for chromosome 5 and chromosome 10 [66]. Three QTLs (qCC-1, qCC-3 and qCC-8) for chlorophyll content in a double haploid (DH) population derived from anther cultures of the Indica and Japonica hybrids ZYQ8/JX17 were detected on chromosomes 1, 3, and 8, respectively [16]. Several QTLs associated with changes in the chlorophyll content of the third leaf were detected on days 13, 16, and 19 after imbibition for 227 rice cultivars from various countries. As a result, it was confirmed that the single nucleotide polymorphism (SNP) cluster at the end of chromosome 11 was significantly associated with the onset of leaf aging [67].

Twenty-six candidate genes related to chlorophyll content were screened in the RM26981-RM287 region using NCBI and RiceXpro. Among candidate genes that have transporter function, *LOC_Os11g37130* is a twin-arginine translocation protein, and is known to be involved in the delta pH-dependent protein transport required for thylakoid membrane formation, in particular [68]. *LOC_Os11g37330* and *LOC_Os11g37660* are Prenyltransferase domain containing proteins, which are known to induce chlorophyll decomposition by enhancing the metabolic flow of α-tocopherol biosynthesis pathway and to be involved in chloroplast biosynthesis [69]. *LOC_Os11g37700* and *LOC_Os11g39020* are ABC transporters in plants and are involved in chlorophyll biosynthesis [70]. Among the genes that have a signaling function, *LOC_Os11g35310* is a phytosulfokine family protein, and encodes proteins related to various chloroplast functions, including photosynthesis, chlorophyll synthesis, carbon fixation, and chloroplast protein synthesis [71]. *LOC_Os11g38260* is a domain found only in photosynthetic organisms, and is a plastid lipid-related protein. It plays a role in chloroplast development, lipid metabolism, and stress response [72]. Among the genes with an RNA binding function, *LOC_Os11g37990* is a CRS1/YhbY domain containing protein and is involved in splicing of group IIA and IIB introns, and *LOC_Os11g37510* is 50S ribosomal protein L4, which is expected to regulate plastid transcription [73,74]. *LOC_Os11g38040* and *LOC_Os11g39140* are Thioredoxin domain 2 containing protein genes with a protein binding function, and serve as a chloroplast redox control system [75]. Among the genes with a nucleotide binding function, *LOC_Os11g37260* is a GTP-binding protein, which controls the chlorophyll content and increases the activity of ROS scavenging enzymes [76]. *LOC_Os11g39540* is a 14-3-3 protein and is expected to play a stay-green-related role [77]. There were five genes with kinase functions. *LOC_Os11g35500* is a kinase-like domain containing protein, and can act as a protein kinase required for the adaptation of the photosynthetic apparatus of the thylakoid membrane [78]. *LOC_Os11g36150*, *LOC_Os11g36160*, *LOC_Os11g36190*, and *LOC_Os11g36200* are expected to be able to control leaf aging by decomposing chlorophyll and controlling plant hormones [79]. Among the genes with a DNA binding function, *LOC_Os11g35320* is a BSD domain containing protein and is expected to be a gene related to leaf aging [80]. *LOC_Os11g36030* is a protein containing HMG-I and HMG-Y domains, and is known to regulate the expression of phytochrome A gene [81]. *LOC_Os11g38870* and *LOC_Os11g39000* are basic helix-loop-helix domain containing proteins, which reduce chlorophyll biosynthesis and are expected to be involved in phytochrome signaling [82]. Among genes with catalytic activity, *LOC_Os11g34210* is a glutamyl-tRNA amidotransferase subunit B and is known to be involved in tRNA-dependent tetrapyrrole biosynthesis [83]. *LOC_Os11g35710* is a terpene synthase family protein and can be involved in the synthesis of the phytol tail of chlorophyll [84]. *LOC_Os11g43200* and *LOC_Os11g43360* are short chain alcohol dehydrogenases and are known to be required for the degradation of chlorophyll b and light harvesting complex II [85]. As a result of comparing the relative expression levels of these 26 chlorophyll related genes, *LOC_Os11g34210*, *LOC_Os11g37130,* and *LOC_Os11g38870* showed a difference in expression level between the CNDH line with high chlorophyll content and CNDH line with low chlorophyll content, and *LOC_Os11g38870* showed the greatest difference. As a result of BLAST analysis of *LOC_Os11g38870* using NCBI, this gene showed a nucleotide sequence similar to that of BHLH062 protein. Basic helix-loop-helix (bHLH) transcription factors are a large family of transcription factors widely distributed in animals and plants and are usually composed of 50–60 amino acids, including a basic region and a helix-loop-helix (HLH) region [86,87]. The bHLH transcription factor can inhibit seed germination, chlorophyll accumulation, and assembly of photosynthetic complexes, and plays important roles in plant growth, development, light signaling, and stress response [82,88]. The *bHLH1* gene induces the expression of stress tolerance-related genes by increasing osmotic pressure, increasing ROS clearance, and enhancing secondary messengers in the stress signaling cascade [89]. It may also improve tolerance to a variety of abiotic stresses by being involved in the regulation of flavonoid synthesis, which plays an important role in ROS homeostasis [90]. It can also increase drought and salt tolerance by increasing the soluble sugar and proline content of plants [91].

*OsbHLHq11* interacts with NAS1, IRO2, OsHRZ1, OsJ_12589, OsJ_32857, OsJ_17424, OS07T0659900-01, OS05T0551000-01, OS04T0444800-02, and OS04T0381700-00. NAS1 (Nicotianamine synthase 1) is a plant Fe chelator involved in metal translocation in plants. Fe deficiency greatly affects chlorophyll synthesis, leading to low yields and poor nutritional quality [92]. IRO2 (Iron-related transcription factor 2) binds to the DNA motif 5′-CACGTGG-3′ in the promoters of Fe deficiency-inducible genes and genes involved in iron homeostasis to prevent iron deficiency and iron absorption from soil [93]. It is a transcriptional activator that contributes to basal tolerance. In particular, it is known to be involved in iron transport during seed maturation and germination. OsHRZ1 is an iron-binding Haemerythrin RING that regulates iron response and accumulation in plants [94]. Os09g0521300 is a TCP transcription factor. It constitutes a plant-specific family of developmental regulators and shares a conserved region predicted to form a bHLH DNA binding domain called the TCP domain [95]. It can be hypothesized that Fe deficiency may simultaneously reduce chlorophyll and lamellar Fe content by delaying the formation of new thylakoids, of which chlorophyll and Fe are integral components, rather than reducing chlorophyll synthesis itself [96].

In this study, the gene *LOC_Os11g38870* (*OsbHLHq11*), which can affect the chlorophyll content related to photosynthesis efficiency, was finally selected through QTL mapping among several candidate genes in the RM26981-RM287 region of chromosome 11. The reason QTL related to photosynthetic efficiency was detected in several chromosomes is presumed to be due to the diverse measured traits and genetic and environmental differences [97]. Therefore, in this study, it is expected that *OsbHLHq11* can increase the photosynthetic efficiency by participating in the chlorophyll content of rice after heading. In addition, *OsbHLHq11* can be utilized as a novel genetic resource for breeding high-yield rice.

## 5. Conclusions

In this study, QTL mapping was performed on leaf color, chlorophyll content, and quantum efficiency of PSII using a 120 Cheongcheong/Nagdong double haploid (CNDH) line to find genes affecting the increase in the photosynthetic efficiency of rice. As a result, QTLs were detected in the RM26981-RM287 region of chromosome 11 for two measurement days, and 26 candidate genes with various molecular functions related to chlorophyll content were selected in this region. *LOC_Os11g38870* (*OsbHLHq11*), which showed the most significant difference, was finally selected by comparing the relative expression level using flag leaf after heading. As a result of BLAST, the sequence of *OsbHLHq11* was very similar to BHLH062 of the *Gramineae* genus, and it was found to interact with proteins involved in Fe homeostasis affecting chlorophyll biosynthesis, such as NAS1, IRO2, and OsHRZ1. Therefore, we propose that *OsbHLHq11* has the potential to increase photosynthetic efficiency by regulating chlorophyll synthesis in rice.

## Figures and Tables

**Figure 1 biology-11-01000-f001:**
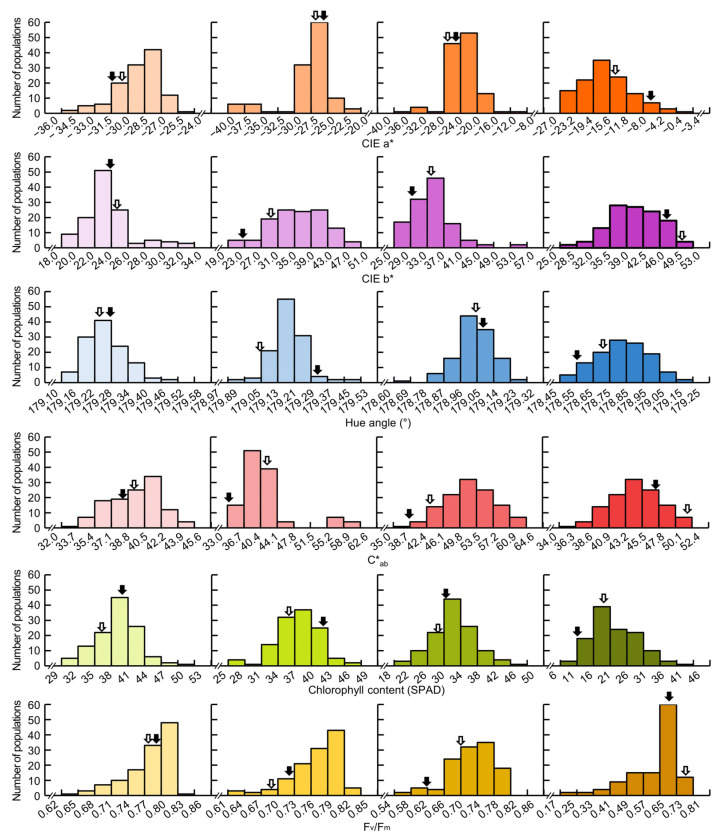
The frequency distribution for CIELAB values, chlorophyll content, and F_v_/F_m_ in 120 Cheongcheong/Nagdong double haploid (CNDH) line. All traits showed normal distribution at 0, 15, 30, and 45 DAH, indicating that the traits are quantitative traits. Black arrow represents Cheongcheong and white arrow represents Nagdong.

**Figure 2 biology-11-01000-f002:**
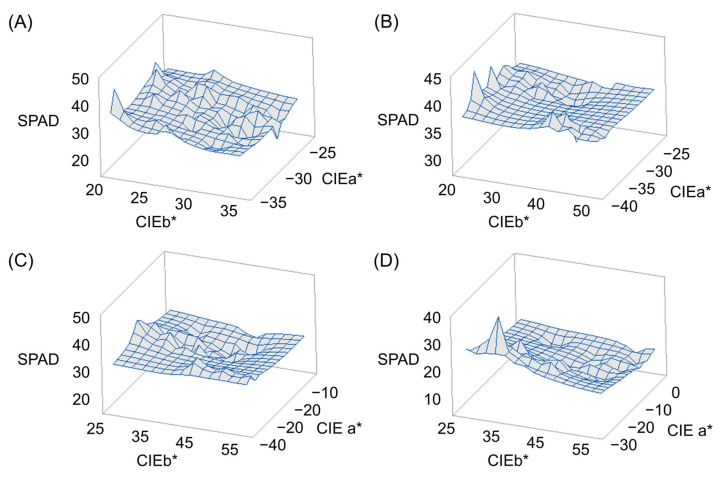
Response surface methodology (RSM) analysis showing relationship of CIEa* and CIEb* on chlorophyll contents. For all measurement days after heading, the lower the CIEa* and CIEb* values, the higher the chlorophyll content: (**A**) 0 days after heading date (DAH); (**B**) 15 DAH; (**C**) 30 DAH; (**D**) 45 DAH.

**Figure 3 biology-11-01000-f003:**
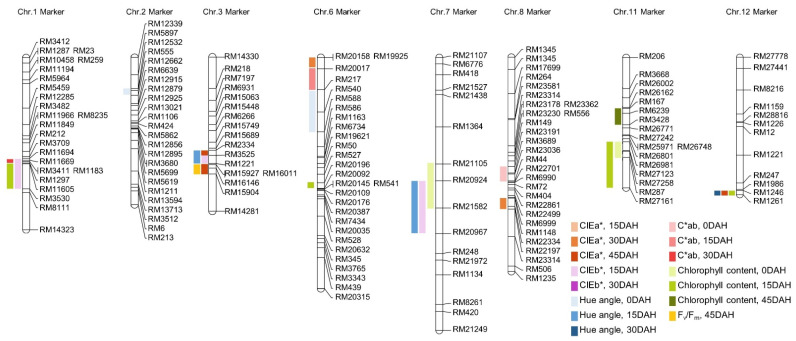
The chromosomal location of QTLs associated with leaf color, chlorophyll content, and F_v_/F_m_ of CNDH linen. The QTLs are shown by bars on the left side of the chromosome. QTLs related to the CIELAB value were detected in chromosome 1, 2, 3, 6, 7, 8, and 12; QTLs related to chlorophyll content were detected in chromosomes 1, 6, 7, 11, and 12; and QTLs related to F_v_/F_m_ were detected in chromosome 3. QTLs above the LOD score ≥ 2.5 were commonly detected in RM26981-RM287 on chromosome 11 and RM1246-RM1261 on chromosome 12.

**Figure 4 biology-11-01000-f004:**
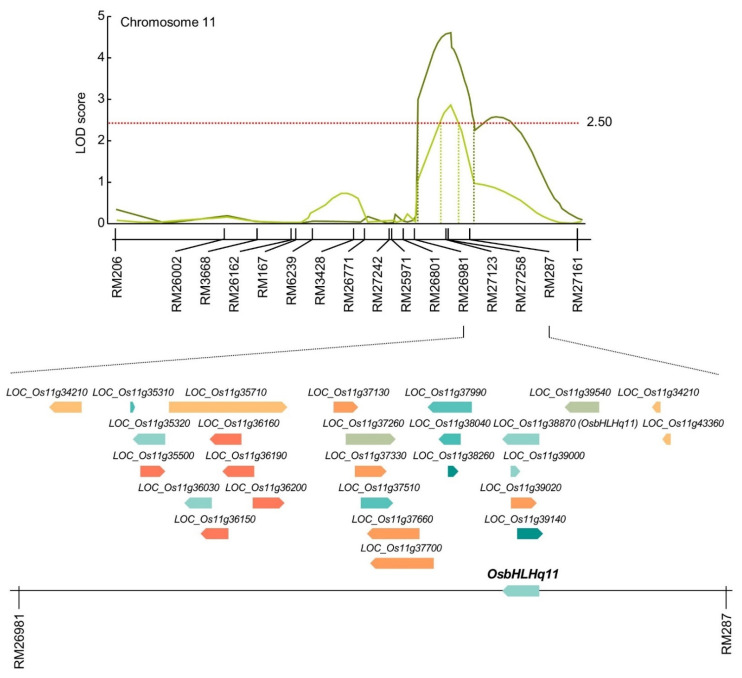
Physical map related to chlorophyll content in rice. In the RM26981-RM287 region on chromosome 11, 2 QTLs associated with chlorophyll content were detected. *OsbHLHq11* is a basic helix-loop-helix transcription factor that can inhibit seed germination, chlorophyll accumulation, and assembly of photosynthetic complexes, and plays important roles in plant growth, development, light signaling, and stress response.

**Figure 5 biology-11-01000-f005:**
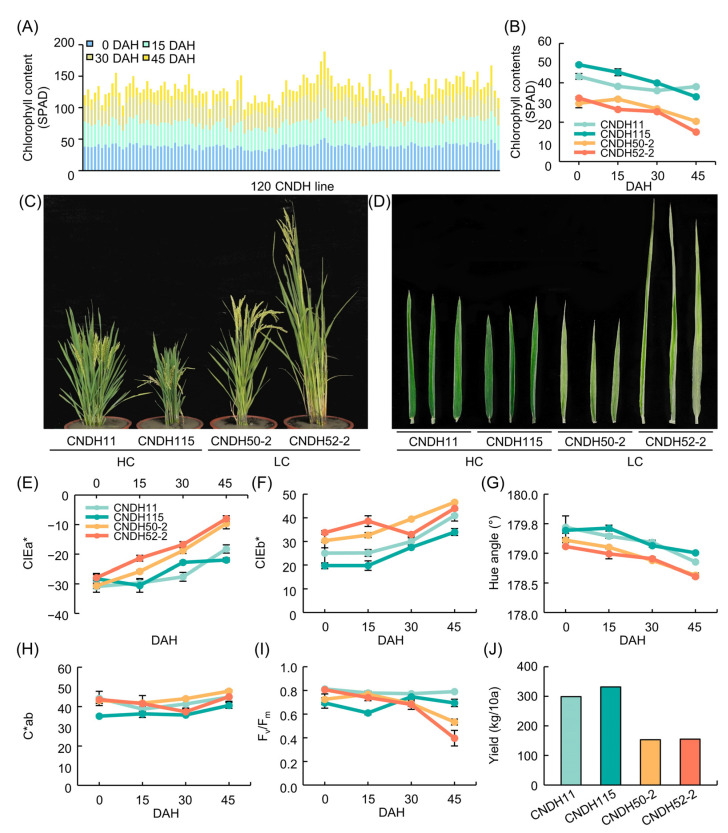
Chlorophyll related trait in CNDH line. (**A**) Changes in chlorophyll content (SPAD) in 120 CNDH line. CNDH11 and CNDH115 had the high chlorophyll content. CNDH50-2 and CNDH 52-2 had the low chlorophyll content. (**B**) Chlorophyll content. (**C**) Phenotypes images. (**D**) Flag leaf images. (**E**) CIEa* value. (**F**) CIEb* value. (**G**) Hue angle. (**H**) C*ab. (**I**) F_v_/F_m_. (**J**) Yield.

**Figure 6 biology-11-01000-f006:**
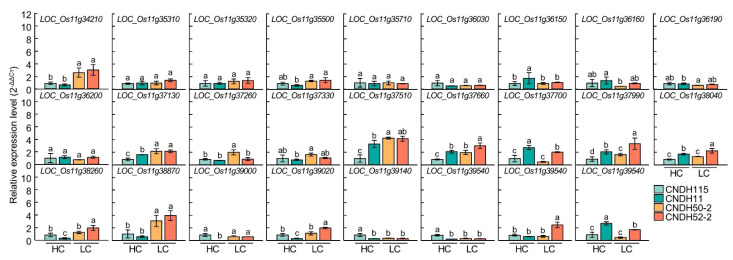
Analysis of relative expression level of 26 chlorophyll-related candidate genes. There were significant differences in *LOC_Os11g34210* and *LOC_Os11g38870.* CNDH115 and CNDH11 are CNDH lines with high chlorophyll content (HC), and CNDH50-2 and CNDH52-2 are the CNDH lines with low chlorophyll content (LC). Means with the same letters were not significantly different by Duncan’s multiple range test at *p* < 0.05.

**Figure 7 biology-11-01000-f007:**
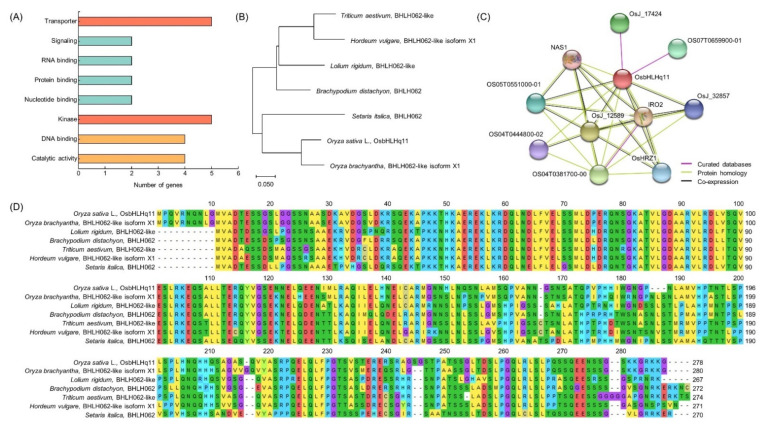
Predicting the function of OsbHLHq11. (**A**) QTL mapping result of chlorophyll content-related gene distribution. (**B**) Protein interaction of OsbHLHq11. (**C**) Phylogenetic tree of *OsbHLHq11* and homology gene. The phylogenetic tree was constructed with 1000 bootstrap replicates. The results of BLAST analysis through NCBI showed that OsbHLHq11 has a very similar sequence to that of BHLH62. Genetic similarity between LOC_Os11g38870 and BHLH62 present in *Triticum dicoccoides, Triticum aestivum, Hordeum vulgare, Zea mays,* and *Setaria italica,* confirmed by phylogenetic tree analysis. OsbHLHq11 belonged to the same group as the *Setaria italica*. (**D**) Multiple sequence alignment of *OsbHLHq11*.

**Table 1 biology-11-01000-t001:** Changes in leaf color, chlorophyll content, and F_v_/F_m_ in Cheongcheong/Nagdong double haploid (CNDH) line.

Plant Traits	DAH ^z^	Parents		DH Line
		Cheongcheong	Nagdong	
CIEL*	0	51.36 ± 2.32 ^y^	50.42 ± 0.99	49.32 ± 3.41
	15	46.34 ± 0.66	51.65 ± 4.32	49.69 ± 3.47
	30	53.86 ± 0.70	53.90 ± 0.99	52.96 ± 4.06
	45	65.66 ± 1.60	64.97 ± 1.10	57.88 ± 7.31
CIEa*	0	−29.90 ± 0.49	−30.01 ± 1.20	−28.90 ± 3.05
	15	−25.85 ± 1.41	−26.16 ± 1.12	−27.84 ± 3.56
	30	−23.79 ± 2.04	−23.79 ± 1.74	−23.78 ± 4.28
	45	−5.53 ± 2.06	−5.53 ± 2.13	−15.73 ± 6.65
CIEb*	0	25.48 ± 2.63	27.54 ± 2.11	26.42 ± 4.02
	15	21.73 ± 2.30	32.71 ± 3.61	29.69 ± 6.22
	30	31.36 ± 2.06	34.77 ± 1.61	34.44 ± 5.85
	45	47.61 ± 2.57	49.16 ± 2.71	40.54 ± 6.64
Hue angle (°)	0	179.28 ± 0.01	179.25 ± 0.04	179.26 ± 0.07
	15	179.31 ± 0.06	179.09 ± 0.03	179.19 ± 0.09
	30	179.08 ± 0.07	179.05 ± 0.03	179.04 ± 0.10
	45	178.54 ± 0.03	178.73 ± 0.02	178.83 ± 0.16
C*ab	0	39.52 ± 0.47	40.99 ± 0.72	39.54 ± 2.67
	15	33.80 ± 1.39	42.06 ± 1.84	41.32 ± 5.98
	30	39.43 ± 0.92	42.49 ± 0.45	41.99 ± 4.84
	45	47.69 ± 1.35	51.94 ± 1.00	44.49 ± 3.45
Chlorophyll contents (SPAD)	0	40.05 ± 0.66	36.55 ± 1.01	39.10 ± 0.92
	15	42.05 ± 1.57	36.38 ± 0.62	37.62 ± 3.91
	30	30.45 ± 0.34	28.10 ± 0.50	32.32 ± 5.08
	45	12.73 ± 0.37	20.15 ± 0.79	22.16 ± 6.86
F_v_/F_m_	0	0.80 ± 0.01	0.80 ± 0.00	0.80 ± 0.04
	15	0.73 ± 0.01	0.78 ± 0.04	0.77 ± 0.04
	30	0.62 ± 0.07	0.71 ± 0.02	0.73 ± 0.05
	45	0.69 ± 0.01	0.74 ± 0.02	0.62 ± 0.12

^z^ DAH means days after the heading date, ^y^ Data are presented as the mean ± standard deviation.

**Table 2 biology-11-01000-t002:** Analysis of correlation between leaf color, chlorophyll content, and F_v_/F_m_ in CNDH line.

DAH ^z^	Plant Traits	Yield	CIEa*	CIEb*	Hue Angle	C*ab	Chlorophyll Content (SPAD)	F_v_/F_m_
0	Yield	1.000						
	CIEa*	−0.025	1.000					
	CIEb*	−0.091	0.042	1.000				
	Hue angle (°)	0.059	−0.491 **	−0.861 **	1.000			
	C*ab	−0.076	−0.540 **	0.784 **	−0.381 **	1.000		
	Chlorophyll content (SPAD)	0.246 **	0.041	−0.596 **	0.489 **	−0.499 **	1.000	
	F_v_/F_m_	−0.126	0.005	0.112	−0.096	0.092	−0.016	1.000
15	Yield	1.000						
	CIEa*	−0.114	1.000					
	CIEb*	0.037	−0.485 **	1.000				
	Hue angle (°)	0.040	−0.213 *	−0.742 **	1.000			
	C*ab	0.074	−0.776 **	0.925 **	−0.437 **	1.000		
	Chlorophyll content (SPAD)	0.303 **	−0.243 **	−0.270 **	0.496 **	−0.078	1.000	
	F_v_/F_m_	0.092	−0.080	0.323 **	−0.335 **	0.262 **	0.017	1.000
30	Yield	1.000						
	CIEa*	0.135	1.000					
	CIEb*	−0.177	−0.077	1.000				
	Hue angle (°)	0.021	−0.693 **	−0.656 **	1.000			
	C*ab	−0.214 *	−0.472 **	0.906 **	−0.288 **	1.000		
	Chlorophyll content (SPAD)	0.273 **	−0.379 **	−0.409 **	0.573 **	−0.211 **	1.000	
	F_v_/F_m_	−0.027	−0.326 **	−0.290 **	0.448 **	−0.136	0.479 **	1.000
45	Yield	1.000						
	CIEa*	−0.005	1.000					
	CIEb*	−0.011	0.725 **	1.000				
	Hue angle (°)	−0.051	−0.816 **	−0.756 **	1.000			
	C*ab	−0.020	0.431 **	0.917 **	−0.506 **	1.000		
	Chlorophyll content (SPAD)	0.032	−0.674 **	−0.668 **	0.644 **	−0.481 **	1.000	
	F_v_/F_m_	−0.057	−0.589 **	−0.300 **	0.443 **	−0.078	0.474 **	1.000

^z^ DAH means days after heading date, * significant at 0.05 level; ** significant at 0.01 level.

**Table 3 biology-11-01000-t003:** QTLs related to the photosynthesis efficiency-related traits of CNDH line.

Plant Traits ^z^	DAH ^z^	QTL ^z^	Chromosome ^z^	Interval Markers ^y^	LOD ^z^	Additive Effect ^x^	*R* ^2 w^	Increasing Effects ^v^
CIEa*	30	qCa6	6	RM20158-RM20017	3.45	−01.31	0.30	Cheongcheong
		qCa8	8	RM1148-RM22197	4.41	−01.52	0.26	Cheongcheong
	45	qCa3	3	RM2334-RM3525	2.63	0–2.89	0.32	Nagdong
		qCa3-1	3	RM15927-RM16146	3.82	−03.08	0.26	Cheongcheong
		qCa12	12	RM1246-RM1261	3.51	−01.96	0.27	Cheongcheong
CIEb*	15	qCb3	3	RM3525-RM1221	2.50	−01.93	0.20	Cheongcheong
		qCb7	7	RM20924-RM20967	3.70	−02.57	0.27	Cheongcheong
		qCb1	1	RM11694-RM3530	2.92	−01.63	0.20	Cheongcheong
Hue angle	0	qHa2	2	RM12532-RM12662	3.63	−00.03	0.26	Cheongcheong
		qHa6	6	RM217-RM588	3.71	0–0.03	0.29	Nagdong
	15	qHa3	3	RM2334-RM1221	4.38	−00.04	0.26	Cheongcheong
		qHa7	7	RM20924-RM20967	2.75	0–0.03	0.23	Nagdong
	30	qHa12	12	RM1246-RM1261	2.80	0–0.03	0.24	Nagdong
C*ab	0	qCab8	8	RM23191-RM44	3.19	0–0.93	0.24	Nagdong
	15	qCab6	6	RM20017-RM217	2.63	0–2.45	0.28	Nagdong
	30	qCab1	1	RM11694-RM1297	3.10	−01.59	0.21	Cheongcheong
Chlorophyll contents	0	qCc7	7	RM21105-RM21582	3.84	0–1.69	0.36	Nagdong
		qCc11	11	RM26981-RM287	2.95	−01.28	0.31	Cheongcheong
	15	qCc1	1	RM11605-RM3530	2.50	−01.35	0.36	Cheongcheong
		qCc6	6	RM20196-RM20092	2.78	0–1.21	0.32	Nagdong
		qCc11-1	11	RM26981-RM287	4.78	−01.66	0.33	Cheongcheong
		qCc11-2	11	RM287-RM27161	2.65	−01.68	0.33	Cheongcheong
		qCc12	12	RM1246-RM1261	2.93	0–1.20	0.32	Nagdong
	45	qCc11-3	11	RM167-RM3428	3.12	0–2.36	0.26	Nagdong
F_v_/F_m_	45	qPqy3	3	RM15927-RM15904	3.45	0–0.06	0.25	Nagdong
Yield	-	qYd6	6	RM20176-RM20387	2.74	−46.19	0.35	Cheongcheong

CIEa*; Ca, CIEb*; Cb, hue angle; Ha, C*ab; Cab, chlorophyll contents; Cc, PSII quantum yield (F_v_/F_m_); Pqy. ^z^ DAH means days after heading date; ^y^ Interval markers are those within the significance threshold on each border of the QTL range; ^x^ The positive values of the additive effect indicate that alleles from Cheongcheong are in the direction of increases in the traits; ^w^ The proportion of evaluated phenotype variations attributable to a particular QTL was estimated using the coefficient of determination (*R*^2^); ^v^ Increased allele is the source of the allele causing an increase in the measured trait.

## Data Availability

Not applicable.

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
