# Peer review of "OsbHLHq11, the Basic Helix-Loop-Helix Transcription Factor, Involved in Regulation of Chlorophyll Content in Rice"

_biology, 2022, doi:10.3390/biology11071000_

Round 1

Reviewer 1 Report

 The author wanted to screen candidate genes that affect photosynthetic efficiency through QTL mapping analysis and predict their function through protein interaction and homology sequence analysis. The results suggest that OsbHLHq11 may be involved in chlorophyll accumulation,enhancing photosynthetic efficiency, which may lead to high yields. However, the author did not carry out experiments to prove the function of osbhlhq11 gene in photosynthesis. Moreover, only the improvement of photosynthetic efficiency does not necessarily increase rice yield, which is related to the distribution of photosynthetic products.

In addition, the results of the manuscript need to be refined and summarized to make it easier for readers to understand.

Reviewer 2 Report

Dear Editor,

Thank you for concerning me as a reviewer of the manuscript submitted to your journal, entitled: "OsbHLHq11, the Basic Helix-Loop-Helix Transcription Factor, Involved in Regulation of Chlorophyll Content in Rice".

In my opinion, the manuscript presented for my assessment is written correctly and the information contained in it is of great cognitive and application significance. Although the work is interesting, I think that the Authors should take a count a very slight modification of this article. I recommend publishing it in "Biology" after a minor revision.

-       the text of the article needs editorial correction (type and font size, spaces, dots….)

Keywords:

Authors should take into account that keywords, according to the rules of writing scientific papers, should not be the same as in the title and usually in alphabetical order.

Introduction

Line 75:  “….. Chlorophyll content is one of the most important physiological properties in rice …” - it is not a rice feature only

Line 84-86: “Chlorophyll affects the photosynthetic capacity of plants such as light blocking, penetration and conversion, which is also linked to crop productivity” - an ambiguous mental shortcut

Line 90: ….leaf photosynthetic rates and chlorophyll content - in reverse order

Materials and Methods

Line 593: Strasser et al. - lack of citation here and in the References part

With best regards,
Monika Tuleja

Reviewer 3 Report

Excellent paper with deep and novel insights related to the potential and the relationship of photosynthesis and yield in rice! I strongly recommend to accept and publish this high quality paper.

Reviewer 4 Report

OsbHLHq11, the Basic Helix-Loop-Helix Transcription Factor, 2

Involved in Regulation of Chlorophyll Content in Rice

Yoon-Hee Jang 1, Jae-Ryoung Park 2, Eun-Gyeong Kim 1 and Kyung-Min Kim 1,* 4

 Q: At what rate are the rice yields declining? And which regions of the World are most likely to be affected? Is this decline specific to a variety?

Q: Is the photosynthetic efficiency the limiting factor, or is the transport system to the sink (Grains) also responsible for declining rice yields?

Q: Is this trait (Yield) regulated by this single gene OsbHLHq11, then to what extent can the yields be enhanced, theoretically?

Q: Grain filling period [Ref 53], Light intensity {Ref 48},  and the day length and environmental stress are equally important factors, which contribute to grains yield. Are these factors influenced by the OsbHLHq11?

Q: Stress regulatory factors should also be discussed in this study.

     Refer to: https://doi.org/10.3390/microorganisms9040774   

Minor issues:

Results and Discussion are too descriptive and loaded with data. The message gets diluted in the process.

References 1 to 10 are not suitable for making a clear presentation of the status of the problem.

 References: Dispense of the old References and the total number may be limited to 40.

Round 2

Reviewer 1 Report

   The author's answer is quite satisfactory. The manuscript has also been comprehensively revised.